# The Synergistic Potential of Combining PD-1/PD-L1 Immune Checkpoint Inhibitors with NOD2 Agonists in Alzheimer’s Disease Treatment

**DOI:** 10.3390/ijms241310905

**Published:** 2023-06-30

**Authors:** Majid Ghareghani, Serge Rivest

**Affiliations:** Neuroscience Laboratory, CHU de Québec Research Centre, Department of Molecular Medicine, Faculty of Medicine, Laval University, Québec City, QC G1V 4G2, Canada; majid.ghareghani.1@ulaval.ca

**Keywords:** Alzheimer’s diseases, dementia, NOD2, PD-1, PD-L1, GSK3β

## Abstract

Our research over the past decade has compellingly demonstrated the potential of Nucleotide-binding oligomerization domain-containing protein 2 (NOD2) receptor agonists in Alzheimer’s disease (AD) treatment. These agonists facilitate the conversation of pro-inflammatory monocytes into patrolling monocytes, leading to the efficient clearance of amyloid-β (Aβ) in the AD-affected cerebrovascular system. This approach surpasses the efficacy of targeting Aβ formation, marking a significant shift in therapeutic strategies. Simultaneously, inhibitors of PD-1/PD-L1 immune check point or glycogen synthase kinase 3 beta (GSK3β), which modulates PD-1, have emerged as potent AD treatment modalities. PD-1 inhibitor exhibits a profound potential in monocytes’ recruitment to the AD-afflicted brain. Recent evidence suggests that an integrated approach, combining the modulation of NOD2 and PD-1, could yield superior outcomes. This innovative combinatorial therapeutic approach leverages the potential of MDP to act as a catalyst for the conversion of inflammatory monocytes into patrolling monocytes, with the subsequent recruitment of these patrolling monocytes into the brain being stimulated by the PD-1 inhibitor. These therapeutic interventions are currently under preclinical investigation by pharmaceutical entities, underscoring the promise they hold. This research advocates for the modulation, rather than suppression, of the innate immune system as a promising pharmacological strategy in AD.

## 1. Introduction

Alzheimer’s disease (AD) is a devastating neurodegenerative disorder characterized by a progressive decline in cognitive function, including memory loss, language difficulties, and changes in behavior. Despite extensive research efforts, AD continues to pose a significant global health burden, with over 50 million cases of dementia worldwide, estimated to rise to over 152 million cases by 2050, and AD accounting for 60–70% of all cases according to the World Health Organization. Furthermore, AD is a leading cause of death worldwide, with a mortality rate of 6.6% per year, and it was the sixth leading cause of death in the United States in 2017. Although the exact etiology of AD remains unclear, certain risk factors have been linked to its onset, including aging, genetics, and lifestyle factors. Aging is the most significant risk factor, with the risk of AD doubling every five years after the age of 65 [1]. Genetics also play a crucial role, with mutations in genes such as APP, PSEN1, and PSEN2 elevating the risk of AD [2]. Furthermore, lifestyle factors, such as diet, physical activity, and social engagement, have been found to be involved in the development of AD [3].

Over the past decades, various theories have been proposed in order to better understand the etiology of AD. The tau hypothesis and the amyloid deposition hypothesis are two well-known theories that elucidate underlying pathogenic mechanisms and may affect treatment options. The tau hypothesis suggests that tau protein is a key component in the maintenance of microtubules, which are essential for neuronal structure and function. Its segregation from microtubules and the development of neurofibrillary tangles is due to tau phosphorylation in AD. This condition, which affects axon transport and neuronal function, leads to the development of cognitive decline in people with AD. On the other hand, the amyloid deposition theory is based on the accumulation of beta-amyloid peptide (Aβ) in senile plaques, which disrupt nerve signaling and induce inflammation, which ultimately leads to neurodegeneration and cognitive decline. BACE1 and the γ-secretase complex are two of the enzymes that degrade amyloid precursor protein (APP) to produce Aβ1-42 (Aβ42), which is the more toxic variant, and Aβ1-40 (Aβ40). Studies have found an association between amyloid deposition and tau hyperphosphorylation, indicating that Aβ synthesis and deposition precede and contribute to tau pathology. The importance of the degree of cognitive decline in the progression of AD is emphasized by the level of oligomeric form of Aβ. Familial AD accounts for approximately 4–6% of cases with an overt genetic predisposition, lending further credence to the amyloid hypothesis. Mutations in the APP, PS1, and PS2 genes are associated with early-onset AD. Mutations in the apolipoprotein E4 (APOE4) gene have been shown to be an important risk factor for late-onset hereditary AD. The amyloid hypothesis is further supported by the fact that these mutations promote amyloid accumulation [4].

Current therapeutic approaches for AD are focused on slowing the progression of the disease and alleviating its symptoms, as a definitive cure remains elusive. Non-pharmacological interventions, such as cognitive training and physical exercise, have also been shown to be effective in enhancing cognitive function and delaying the onset of dementia. Moreover, targeting systemic innate immune cells, such as monocytes and macrophages, has emerged as a potential therapeutic avenue for AD [5]. Monocytes are a type of white blood cell that play important roles in the innate immune response and are generated in the bone marrow from hematopoietic stem cells and differentiate into distinct subsets based on the expression of specific cell surface markers. However, there are some differences in the markers that are expressed on monocytes in mice and humans. For example, in mice, the main subsets of monocytes are Ly6C^high^ (equivalent to human classical/inflammatory monocyte) and Ly6C^low^ (equivalent to human non-classical/patrolling monocyte), whereas in humans, the main subsets are CD14^++^CD16^−^ (classical) and CD14^+^CD16^+^ (non-classical) monocytes. Once the monocytes are generated, they need to exit from bone marrow and carucate in blood to migrate to tissues into macrophages. In terms of markers involved in monocyte migration, CCR2 is important for the migration of both monocytes from the bone marrow to the bloodstream and to sites of inflammation. Markers involved in monocyte differentiation and activation, such as NF-kB, NR4A1, and NOD2, are also expressed on both human and mouse monocytes. However, there may be some differences in the specific signaling pathways and downstream effects of these markers in mice and humans. The proliferation, differentiation, and migration of monocytes has been already reviewed in our previously published paper [6].

Over the last decade, the studies of our laboratory uncovered the crucial role of Nucleotide-binding oligomerization domain-containing protein 2 (NOD2) in the function of monocytes. Indeed, we showed the activation of NOD2 receptors in the function of monocytes as a key switch for monocyte conversation from classical to patrolling monocytes in murine models of AD [6,7], as well as autoimmune diseases of multiple sclerosis (MS) [8], which contribute to the clearance of toxic amyloid beta (also known as Aβ), a hallmark of AD, and myelin debris in MS. The switch of monocyte conversation from classical to patrolling monocytes in response to NOD2 signaling was shown to play a powerful beneficial role in engulfing the toxic Aβ from the luminal side of the cerebrovascular system, which finally leads to improvement in AD symptoms. 

Building on our laboratory’s research focus on utilizing monocytes to clear Aβ from cerebral blood vessels, our paper aims to integrate the role of monocytes in AD and propose strategies to enhance the efficacy of ongoing therapeutic trends. Specifically, we will discuss NOD2 and its role in AD and explain how the PD-1/PD-L1 immune checkpoint is involved in AD. We will also address how NOD2 activation leads to the activation of monocyte conversion, and we will propose the hypothesis that NOD2 activation and PD-L1 inhibitors may complement current treatments and improve outcomes for AD patients.

In conclusion, our paper seeks to shed light on the vital role of monocytes in AD and to propose novel therapeutic approaches that may prove effective in slowing or reversing the progression of the disease. By leveraging the latest advances in our understanding of the immune system and molecular mechanisms underlying AD, we hope to contribute to the ongoing efforts to combat this devastating disorder.

## 2. NOD2 in Alzheimer’s Diseases

NOD2 receptors are a type of receptor that is involved in the body’s innate immune response. They are located within certain immune cells and their main function is to recognize and respond to bacterial peptidoglycan (PGN) fragments, which are typically found in bacterial cell walls. This recognition leads to the activation of signaling proteins, which stimulate the production of pro-inflammatory cytokines and other immune mediators that help to fight off bacterial infections. NOD2 receptors also play a role in autophagy, a cellular process that helps to break down and recycle components of the cell. NOD2 receptors are expressed in various immune cells, including bone marrow cells and monocytes [9]. Bone marrow cells, which are progenitor cells for all blood cells, express NOD2 receptors at various stages of their differentiation. Studies have shown that NOD2 signaling plays a role in the differentiation of bone marrow cells into monocytes and dendritic cells, two types of immune cells that play a critical role in the innate immune response [10].

Our previous studies showed that activation of NOD2 by muramyl dipeptide (MDP), a small molecule that is derived from PGN, converts inflammatory monocytes into patrolling monocytes, which have a greater ability to phagocytose and clear vascular amyloid associated with AD [10]. Patrolling monocytes are a subpopulation of monocytes that circulate in the bloodstream and are capable of migrating to various tissues to perform immune surveillance functions. Recent studies have shown that patrolling monocytes have a greater ability to phagocytose compared to classic monocytes or brain residential microglia. One study conducted by Auffray et al. (2009) showed that patrolling monocytes were more effective at phagocytosing apoptotic cells than other monocyte subsets. Indeed, they found that patrolling monocytes expressed higher levels of the scavenger receptor CD163, which is known to play a role in phagocytosis of apoptotic cells [11]. Another study by Grabert et al. (2016) investigated the role of patrolling monocytes in the phagocytosis of Aβ. They demonstrated that patrolling monocytes were the primary phagocytic cells responsible for clearing Aβ from the brain, suggesting that these cells may play an important role in the pathogenesis of AD [12]. These studies were recently reviewed by us [6]. Taken together, these studies suggest that patrolling monocytes have a unique ability to phagocytose cellular debris and foreign material in a variety of tissues, including the brain vascular system. This may have important implications for the development of new therapies for diseases that involve excessive accumulation of toxic proteins and cellular debris, such as AD and MS, respectively. 

The conversation of monocyte subsets seems to be dependent on NR4A1 (also named Nur77), a transcription factor that plays a crucial role in the production of patrolling monocytes [13]. The recruitment of patrolling monocytes to vascular amyloid deposits is still not fully understood, but it is believed that CCR2, CCR5, and CX3CR1 each are partially involved in their recruitment [14,15]. Additionally, the intravital two-photon microscopy of our laboratory has demonstrated that patrolling monocytes are specifically attracted to the luminal wall of Aβ-positive vasculatures, and their removal has been shown to lead to a significant decrease in Aβ plaques [6]. This suggests that increasing the number of patrolling monocytes in the vasculature could be a potential therapeutic avenue for AD. The mechanism of Aβ efflux from the brain to the vasculature, called the sink effect, is also being explored as a potential avenue for AD therapy [6].

The previous studies of the laboratory uncovered a high potential of the NOD2 ligand, MDP in monocyte conversation, and pathophysiology of AD in its animal models. Indeed, we showed that MDP triggers the conversion of inflammatory monocytes (Ly6C^high^) into patrolling monocytes (Ly6C^low^) in an NOD2-dependent manner. Indeed, mice injected with MDP had a strong increase in the number of patrolling monocytes associated with a decrease in inflammatory monocytes. However, when NOD2^−/−^ mice were injected with MDP, no switch was observed, indicating that MDP triggers the conversion in an NOD2-dependent manner. The results also showed that MDP has a powerful impact on the short-term memory of APP mice, an excellent model of AD for studying the amyloidopathy, since mice display cognitive impairment at 6 months of age. The principal manifestation of this decline is short-term memory loss and cerebral amyloid angiopathy [10,16]. In another study, we also found that MDP-stimulated NOD2 was associated with the increased expression of the Aβ transporter, the low-density lipoprotein receptor-related protein-1 (LRP-1). LRP-1 transports Aβ from the abluminal to luminal side of the blood brain barrier (BBB) where patrolling monocytes can scavenge it [6]. In sum up, using the potential of patrolling monocytes for engulfing cerebrovascular Aβ instead of using approaches that target Aβ plaque formation was found to be a better strategy in AD treatment.

As an auxiliary elucidation to enhance the portrayal of our methodology in scrutinizing the behavior of monocytes in Aβ elimination from the brain and cerebrovascular system, we deemed it beneficial to elucidate that we employed chimeric mice in our study to distinguish between brain-resident microglia and bone-marrow-derived (peripheral) macrophages within the CNS. Adult APP mice, serving as models of AD, had their bone marrow (BM) hematopoietic cells supplanted with green fluorescent protein (GFP)-expressing cells, facilitating the identification of infiltrating monocytes or bone-derived macrophages via the presence of GFP. In contrast, GFP-negative macrophages were identified as microglial cells that had differentiated into macrophages. The BM utilized for this purpose was procured from sex-matched CX3CR1-GFP donors and transplanted into adult male APP recipient mice. These recipient mice had their BM hematopoietic cells previously eradicated through chemotherapy. Approximately 8 weeks post-transplantation, to authenticate the successful chimerism, we evaluated peripheral blood samples via Fluorescence Activated Cell Sorting (FACS) to quantify the number of donor-derived cells in the recipient mice [16].

In the context of Aβ removal from the cerebrovascular system, we employed the APP mouse model to scrutinize the activity of monocytes within the vasculature. Aβ was stained with Congo red, which was injected into the cerebral spinal fluid via the cisterna magna, prior to imaging sessions. Immediately before imaging, Qdot 705 was administered via the tail vein to label the blood vessels. Given that we had generated chimeric mice in which the APP mice’s BM was replaced with GFP-tagged CX3CR1 cells, we were able to monitor the monocyte behavior inside the vessels using in vivo imaging [16].

## 3. The General Role of PD-1 and PD-L1 in Immune System

Programmed cell death protein 1 (PD-1) is a protein found on the surface of some immune cells, such as T cells, B cells, and natural killer cells. Its function is to regulate the immune response by interacting with another protein called programmed death ligand 1 (PD-L1) or PD-L2. PD-L2 has three times higher affinity for PD-1 than PD-L1. Although both PD-L1 and PD-L2 are involved in immune checkpoint inhibition, PD-L1 is of more clinical significance because it is more widely expressed in different tissues [17]. This ubiquitous expression of PD-L1 makes it a target for immunotherapy drugs that block the interaction between PD-L1 and PD-1. The interaction between PD-1 and PD-L1 is an important mechanism for maintaining immune system homeostasis, preventing the immune system from attacking healthy cells in the body and causing autoimmune diseases. When PD-1 binds to PD-L1, it sends a signal to immune cells to stop attacking PD-L1-expressing cells, a process known as “immune checkpoint inhibition”. The interaction between PD-1 and PD-L1 plays an important role in regulating the immune system’s response to infectious agents, tumors, and tissue damage [18]. When an immune cell recognizes a target, such as a tumor cell, it undergoes activation and proliferation, leading to an immune response against the target. However, prolonged activation of immune cells can lead to tissue damage and autoimmune diseases. The interaction between PD-1 and PD-L1 helps prevent this by inhibiting the immune response. Expression of PD-L1 on cancer cells and other abnormal cells may help them evade the immune system by suppressing the immune response. In addition, cancer cells can also upregulate PD-L1 expression in response to invasion of the immune system. By expressing PD-L1, cancer cells can “turn off” immune cells that would normally recognize and effectively kill them. This mechanism is called immune evasion. Blocking the PD-1/PD-L1 pathway with drugs has been shown to be an effective treatment for certain types of cancer because it allows the immune system to attack and destroy cancer cells. This treatment is called immune checkpoint therapy [19]. The role of this immune checkpoint in neuroinflammation is discussed in depth in a recently published article [20].

## 4. PD1/PD-L1 in AD and Its Murine Models

Bianchini et al. studied PD-L1 and PD-L2 expressions on both classical (inflammatory Ly6C^high^) and non-classical (patrolling Ly6C^low^) monocytes. They observed that classical monocytes did not express PD-L1, while non-classical monocytes did. None of the monocytes expressed PD-L2. This difference allowed them to discriminate between the two subsets in vivo and track and monitor patrolling monocytes and their interactions [17]. In two studies conducted in 2016 and 2019, Michal Schwartz’s team reported a potential therapeutic strategy of immune checkpoint blockade targeting the PD-1 pathway for AD. Using mouse models of AD, they demonstrated that PD-1 blockade, achieved through injection of a PD-1-specific blocking antibody, elicited an interferon (IFN)-γ-dependent systemic immune response. This response led to the recruitment of monocyte-derived macrophages to the brain, which were examined 7 and 14 days after the first antibody injection. Furthermore, this immune response resulted in the clearance of cerebral Aβ plaques and improved cognitive performance in mice with established AD pathology. The authors further supported their findings using another AD model, APP/PS1 mice at 8 and 15 months of age. They observed a reduced hippocampal Aβ accumulation following PD-1 blockade [21,22].

In contrast to the systemic injection of a PD-1 blocker, which demonstrated a beneficial role in AD, Kummer and colleagues in 2021 investigated the role of PD-1 and its ligand PD-L1 in AD pathology. They found that loss of PD-1 increased Aβ plaque accumulation and decreased microglial Aβ uptake in APP/PS1 mice. According to their study, PD-L1 levels were higher in the cerebrospinal fluid (CSF) of AD patients compared to that of control patients. The source of PD-L1 was identified as astrocytes located near Aβ in the brains of AD patients but not case controls, and this finding was confirmed in the APP/PS1 mouse AD model. PD-L1 upregulation was considered an initial phenomenon, as it was present even in mice with very few amyloid plaques as young as 4 months of age. Under normal conditions, astrocytes do not display detectable levels of PD-L1; however, its expression is induced upon stimulation with synthetic Aβ1-42 or a mixture of inflammatory mediators, including IFN-γ and tumor necrosis factor-α (TNF-α). They proposed that this increased expression of PD-L1 by astrocytes could explain the elevated levels of PD-L1 in the CSF of patients with AD. In addition, they found increased expression of PD-1 in microglia of patients and APP/PS1 models located near Aβ plaques. Using PD-1-deficient mice in AD models, they demonstrated promotion of TNF-α expression and exacerbation of cognitive behavior following treatment with synthetic Aβ1-42 [23].

In 2022, Chuang-Rung Chang and colleagues performed a study that revealed interesting insights into the expression levels of PD-1 and PD-L1 in different subsets of T cells from AD patients. Although PD-1 expression levels in certain groups of T cells were similar between patients with AD and healthy individuals, there was only higher expression in CD4^+^ CD25^+^ T cells in AD patients. They found a significant upregulation of PD-L1 in variant T cell populations. Specifically, there was a higher expression of PD-L1 on CD8^+^, CD3^+^CD56^+^, CD4^+^, and CD4^+^CD25^+^ T cells in AD patients. To better understand the correlation between the expression patterns of PD-1/PD-L1 and the progression of AD, the team classified the patients into mild, moderate, and severe stages based on the Clinical Dementia Rating. Interestingly, they found that PD-L1 expression levels tended to increase with AD progression. Notably, there was a significant upregulation of PD-L1 in CD3^+^CD56^+^ and CD4^+^CD25^+^ T cells in patients with mild AD, and in CD4^+^ and CD8^+^ T cells in patients with moderate AD. In addition, the team observed a significant increase in the number of PD-L1^+^CD4^+^/CD8^+^ T cells in patients with moderate AD compared with healthy subjects [24].

Beneficial results were found in the study examining the impact of systemic injection of a PD-1 inhibitor on AD pathogenesis [21,22]. However, another study utilizing PD-1-deficient mice to examine the impact of microglial PD-1 deletion on AD pathogenesis found that this loss resulted in increased neuroinflammation and exacerbated AD symptoms [23]. This distinction shows that elimination of PD-1 may not be a helpful treatment strategy and that PD-1 may be engaged in additional signaling pathways that are undiscovered but essential for healthy immune system function. Instead, the systemic injection approach for blocking PD-1 and PD-L1 interaction, which has already shown potential in suppressing PD-1 and PD-L1 interaction, may be a better choice for further research. It is worth noting that Michal Schwartz’s team used a PD-1 blocker antibody in their AD studies, the same molecule targeted by Merck & Co Inc.’s Keytruda (pembrolizumab), a medication that received FDA approval in early 2023 for cancer treatment [25]. 

It is essential to highlight that the seemingly contradictory results of these studies could be attributed to the different methods used to modulate the PD-1 pathway. While ablation of microglial PD-1 may have broader effects on the immune system that may potentially counteract the desired advantages, systemic injection of a PD-1 blocker intends to disrupt the interaction between PD-1 and its ligands, exclusively targeting the PD-1 pathway. Determining the molecular mechanisms and signaling pathways involved in PD-1 and PD-L1 regulation, as well as their impacts on the immune system and AD progression, is therefore vital for future research.

Moreover, it is essential to explore the long-term effects of systemic PD-1 blockade as a potential therapeutic strategy for AD, as well as its safety and efficacy in human patients. Given that immune checkpoint inhibitors targeting the PD-1 pathway have already been used in cancer therapies, there is potential to leverage existing knowledge and clinical experience to accelerate the development of AD treatments.

Classical monocytes, intrinsic to the initial inflammatory response, can differentiate into tissue-residing macrophages, thereby enhancing inflammation and influencing the course of chronic diseases. This transmutation is orchestrated through the interactive dialogue between CCR2, a monocyte-associated receptor, and CCL2, a chemokine functioning as a homing signal for monocytes at inflammation sites [26]. Conversely, non-classical monocytes typically exhibit anti-inflammatory tendencies, reinforcing vascular homeostasis. These monocytes, devoid of CCR2 unlike their classical counterparts, evolve into macrophages via the concerted operation of CCL3, a distinct chemokine, and CX3CR1, a monocyte-bound receptor. Nevertheless, the intricacies and implications of this phenomena are yet to be comprehensively discerned and constitute an active area of exploration [27,28].

With regard to AD, PD-1 blockade escalated the prevalence of CD45^high^CD11b^+^ cells within the murine brain, effectively augmenting the influx of monocyte-derived macrophages. These cells, a subtype of myeloid cells encompassing monocytes and macrophages, are denoted by their prolific expression of lymphocyte antigen 6c (Ly6C) and the chemokine receptor CCR2, signifying their potential origin as infiltrating monocytes morphed into macrophages. Further investigation pointed towards the involvement of a specific monocyte subset characterized by an abundant expression of Ly6C and CCR2. Classical monocytes, exhibiting a pronounced presence of CCR2, which plays a pivotal role in the emigration of monocytes from bone marrow to the blood, were found to increase upon PD-1 inhibition, leading to an augmentation in circulating CCR2+ monocytes and regulatory T cells (Tregs). Notably, when CCR2 was blocked, the PD-1 inhibitor’s ability to promote monocyte recruitment to the brain and Aβ elimination diminished significantly, while the accumulation of Tregs in the brain was substantially reduced [21,22,29,30].

Progressing with these investigations, Schwartz’s team employed single-cell RNA sequencing to characterize the infiltrating cells, deducing that the incoming monocyte-derived macrophages displayed heterogeneity and likely comprised various activation states. These macrophages were discovered to express a number of molecules potentially facilitating disease modification, including macrophage scavenger receptor 1 (Msr1), insulin-like growth factor-1 (igf1), lymphatic endothelium-specific hyaluronan receptor (lyve1), and stabilin-1 (Stab-1), hallmarks of anti-inflammatory macrophages. Subsequently, they engaged MSR1-deficient mice to assess the impact of the PD-1 inhibitor in AD, revealing that these mice lacked the capacity to react to PD-L1 blocking antibody and demonstrated no improvement in cognitive function, suggesting an indispensable role of MSR1-expressing macrophages in the reparative process [21,22,29,30].

In summary, while PD-1 blockade in AD amplifies monocyte influx into the brain, MDP-activated NOD2 appears to shift the balance from classical to non-classical monocytes, concurrently enhancing monocyte recruitment. Conversely, PD-1 inhibition bolsters recruitment in a CCR2-dependent fashion, given CCR2’s crucial role in monocyte transmigration from bone marrow to the bloodstream. It was demonstrated that the augmented monocytes in AD brains originate from peripheral monocytes, but the exact proportion of classical and non-classical monocytes in such infiltration in response to PD-1 inhibition was not investigated. Despite this, the brain-resident macrophages were found to express several markers with anti-inflammatory functions. The impacts of the NOD2 agonist MDP and the PD-1 inhibitor on monocyte transformation and recruitment to the AD brain are diagrammatically illustrated in Figure 1; however, certain phenomena remain unexplored and necessitate further elucidation.

## 5. Glycogen Synthase Kinase 3 Beta (GSK3β) Role in AD Pathology

Glycogen synthase kinase 3 beta (GSK3) is a serine/threonine kinase that is essential for a variety of cellular functions, such as glycogen metabolism, cell division, proliferation, and apoptosis. The development of various neurodegenerative diseases, including AD, has been connected to GSK3 dysregulation. Due to its role in controlling the two primary pathological manifestations of AD, neurofibrillary tangles and amyloid plaques, GSK3 is linked to the onset and course of the disease [31]. Neurofibrillary tangles, one of the hallmarks of AD pathology, consist of abnormal clumps of tau protein, which is a microtubule-associated protein essential for stabilizing the cytoskeleton in neurons. In AD, tau protein becomes hyperphosphorylated, resulting in its detachment from microtubules and its aggregation into neurofibrillary tangles. GSK3β is a crucial kinase that phosphorylates tau protein, and its dysregulation has been shown to contribute to the development of neurofibrillary tangles [32]. Specifically, GSK3β can phosphorylate tau at multiple sites, including Ser202, Thr205, and Ser396, which are critical for tau aggregation and neurofibrillary tangle formation [31]. The other hallmark of AD pathology, amyloid plaques, are composed of abnormal deposits of Aβ peptides derived from the processing of amyloid precursor protein (APP). GSK3β is involved in the processing of APP, resulting in the production of Aβ peptides. GSK3β phosphorylates APP at Ser655, which promotes its cleavage by beta-secretase and the production of Aβ peptides. Furthermore, GSK3β has been shown to be involved in regulating Aβ clearance by interacting with the Aβ-degrading enzyme neprilysin.

Apart from its role in regulating neurofibrillary tangles and Aβ plaques, GSK3β has been implicated in regulating other processes that contribute to the pathogenesis of AD, including oxidative stress via the Nrf2-ARE pathway, inflammation, and synaptic dysfunction [33]. GSK3β can regulate the expression and activity of various transcription factors, such as NF-κB, which are involved in regulating inflammatory processes. Additionally, GSK3β can modulate synaptic plasticity [34] and function by regulating the phosphorylation of various synaptic proteins, including synapsin I and PSD-95 [35,36].

## 6. GSK3β Has Been Implicated in AD in Conjunction with the PD-1/PD-L1 Signaling Pathway

Although the relationship between GSK3β and PD-1/PD-L1 signaling is not completely understood, some studies suggest that GSK3β could be involved in the regulation of PD-L1 expression and function. Regulatory T cells express high levels of PD-1 and are essential for preventing autoimmunity by suppressing the activation of self-reactive T cells. Inhibition of GSK3β activity can increase PD-1 expression on regulatory T cells and enhance their suppressive function, which could be advantageous in the treatment of autoimmune diseases [37,38]. For instance, Taylor et al. [39] examined the effect of GSK3 using GSK-3 siRNA or SB415286, in CD8(+) T cells, which was found to be involved in AD pathology [40], and they found a huge decrease in PD-1 expression. It is interesting to note that there was no change in the expression level of intracellular IL-2, which is also involved in the improvement of AD [41]. They concluded that GSK3 plays a role as a key upstream kinase that regulates PD-1 expression in CD8(+) T cells [39,42]. Another study reported another connection between GSK3 and PD-1 suggesting that GSK3β may act directly downstream of the PD1/PDL1 axis, after observing that the PD1-blocking antibody decreased the tau hyperphosphorylation and GSK3β activity. In this study, the team of Zhou Chen found that PD-1 and PD-L1 levels were increased in the hippocampus of APP/PS1 and the cortex and hippocampus of 5×FAD models of AD. Indeed, this increase in their expression was also observed following intracerebroventricular injection of Aβ to mice. Moreover, the increased levels of PD-1 and PD-L1 following Aβ injection were associated with decreased levels of p-GSK3β (Ser9), a form of GSK3β that is inactive and unable to phosphorylate its targets. In contrast, activated GSK3β showed a direct positive correlation with PD-1 expression in AD models [43]. Furthermore, Zhang et al. utilized lithium, an FDA-approved GSK3β inhibitor, to suppress GSK3β activity, and found that it reduced the severity of experimental autoimmune encephalomyelitis (EAE), a mouse model of MS, by promoting the function of regulatory T cells [44]. GSK3β-selective inhibition has also been shown to play anti-inflammatory and anti-oxidative roles for microglia. This was found to be a nuclear factor erythroid 2-related factor 2 (Nrf2)-independent response of microglia, the principal regulator of redox homeostasis [45].

To summarize, dysregulation of GSK3β has been implicated in the development and progression of AD through its involvement in the regulation of multiple processes that contribute to the pathogenesis of the disease. Therefore, GSK3β inhibitors may have promising therapeutic applications in the treatment of AD by targeting the underlying pathological processes of the disease. In accordance with this notion, a retrospective cohort study was conducted in 2022 to assess the association between lithium chloride use and the incidence of dementia and its subtypes. The study identified 548 patients who were exposed to lithium chloride and 29,070 patients who were not exposed to lithium chloride over a period of 15 years. It is worth noting that lithium chloride is the only FDA-approved GSK3 inhibitor that is prescribed for epilepsy and bipolar disorder. These findings revealed that lithium chloride use was associated with a lower risk of receiving a diagnosis of dementia, including a lower risk of being diagnosed with either AD or vascular dementia [46]. These results suggest that GSK3β inhibition may be a viable therapeutic approach for AD, and further studies are warranted to explore the potential of other GSK3β inhibitors in the treatment of this devastating disease.

A rudimentary depiction of GSK3β’s involvement in the expression of PD-1 and the phosphorylation of Tau is elucidated in Figure 2. Nonetheless, this preliminary understanding warrants additional corroboration.

## 7. Conclusions

Drawing upon the recent advancements in understanding the potential of engaging systemic innate immune cells, such as monocytes, our research lends substantial weight to the proposition that such interventions could serve as a potent strategy to counteract aging-related AD. Our exploration illuminates the pivotal role of NOD2 activation in augmenting the population and proficiency of patrolling monocytes, thereby bolstering the clearance of Aβ along the luminal facet of the cerebral and periphery vasculatures. This revelation paves the way for a promising therapeutic pathway in the management of AD.

While each of these therapeutic strategies demonstrates remarkable efficacy in AD treatment independently, our research intimates that an integrated approach could yield superior outcomes. The PD-1 inhibitor exhibits a profound potential in marshalling monocytes to the brain of an AD patient, which appears to concurrently augment anti-inflammatory macrophages. Meanwhile, the stimulation of NOD2, via MDP, facilitates the conversion of inflammatory monocytes into patrolling monocytes. There is compelling evidence to suggest that MDP could act as a catalyst for this monocyte conversation, and the subsequent recruitment of patrolling monocytes into the brain is stimulated by the PD-1 inhibitor.

Consequently, we advocate that our synergistic strategy could emerge as a pivotal instrument for the development of novel AD therapies, thereby enhancing patient prognosis and their quality of life. A lucid depiction of this combined therapeutic approach is encapsulated in Figure 3.

## Figures and Tables

**Figure 1 ijms-24-10905-f001:**
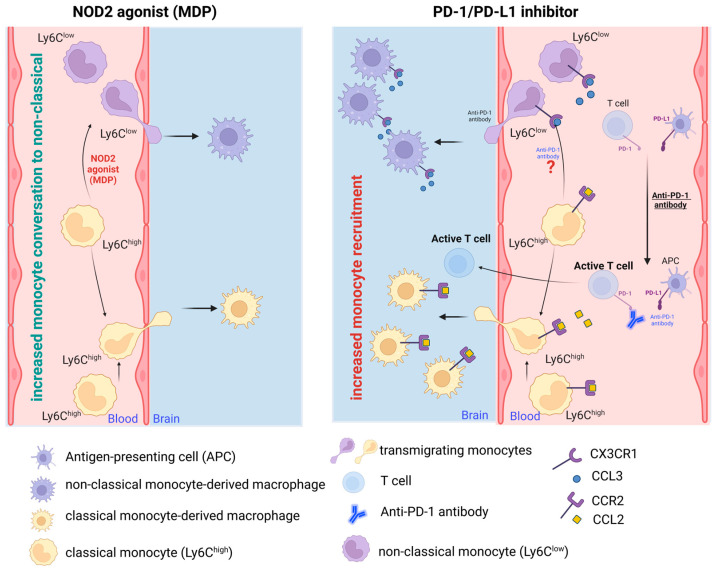
**A detailed graphical representation of the process of monocyte mobilization to the brain and their subsequent transformation.** NOD2 agonists, such as MDP, incite in circulation a conversion of classical monocytes (Ly6C^high^) into their non-classical counterparts (Ly6C^low^), without significant impact on monocytes’ recruitment to the brain. In other words, if there is a potential low infiltration of monocytes to the brain, MDP causes these cells to be more of a patrolling subset. On the other hand, the inhibition of PD-1/PD-L1, exemplified by the use of an anti-PD-1 antibody, is demonstrated to escalate the monocytes’ migration to the brain. The specific impact of this inhibition on each subpopulation of classical and patrolling monocytes has not been conclusively elucidated, although the presence of certain anti-inflammatory markers has been identified on monocyte-derived macrophages. Therefore, the effects of an anti-PD-1 antibody on the conversion of monocytes warrant further examination. Pertaining to T cell activation, the therapeutic application of an anti-PD-1 antibody obstructs the interaction between PD-1 and PD-L1, subsequently triggering T cell activation, which is found to increase the monocyte recruitment to the brain.

**Figure 2 ijms-24-10905-f002:**
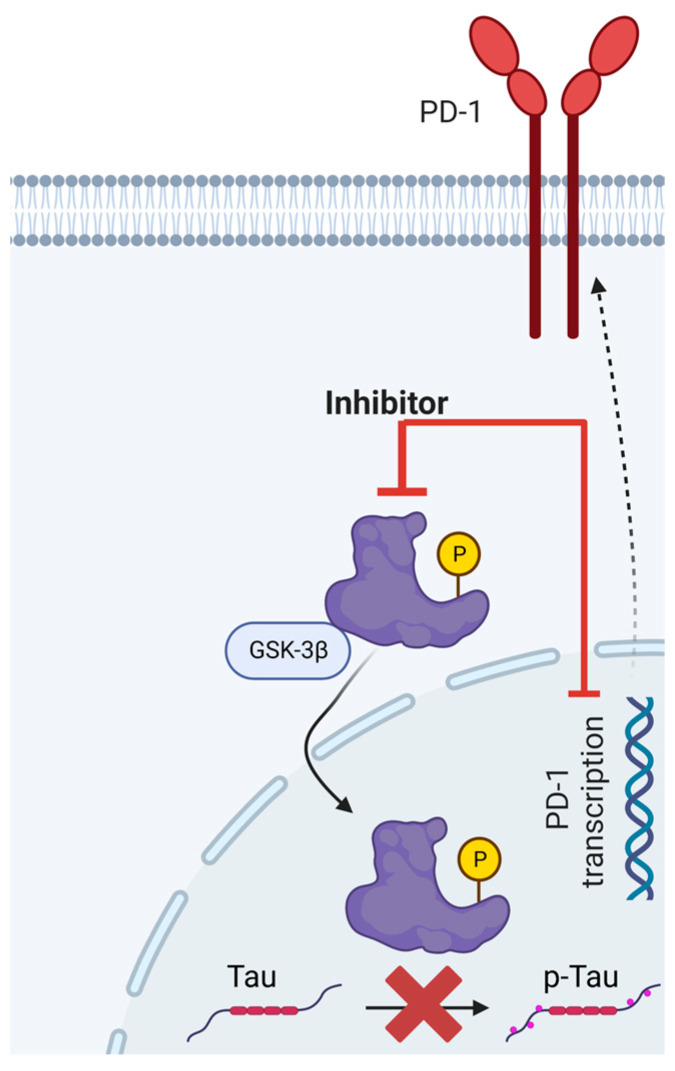
**An uncomplicated visual representation of the instrumental function of GSK3β in PD-1 expression and tauopathy.** Blockers of GSK3β curtail the action of the activated, phosphorylated variant of GSK3β, which appears to participate in the hyperphosphorylation of the tau protein. This intricate process precipitates the tau protein’s dissociation from microtubules and its subsequent congregation into neurofibrillary tangles. Moreover, it has been ascertained that inhibitors of GSK3β suppress the transcriptional activity of PD-1, subsequently catalyzing a decline in PD-1 expression.

**Figure 3 ijms-24-10905-f003:**
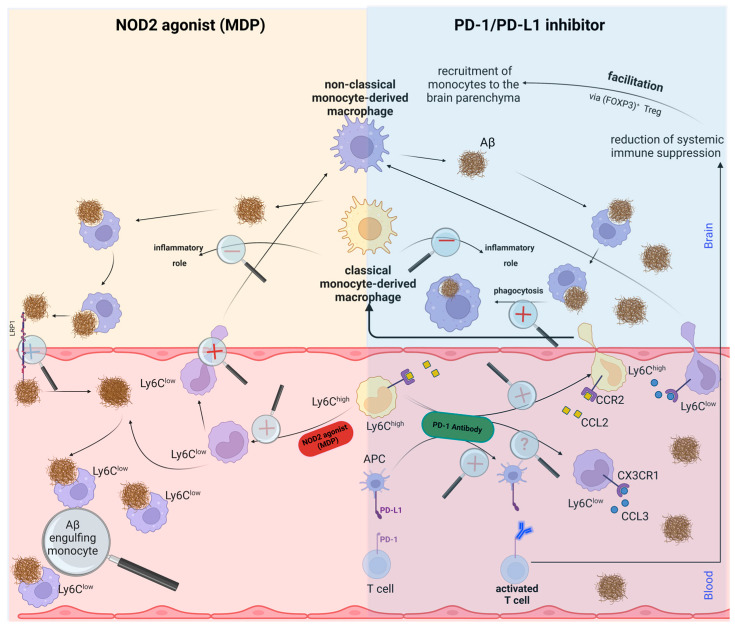
**Diagrammatic elucidation juxtaposing NOD2 agonist therapy and PD-1 inhibitor therapy, culminating in a synthesis of both treatment strategies.** The upper left quadrant of the graphical précis delineates the process by which the administration of a NOD2 agonist instigates the transformation of inflammatory monocytes into vigilant, patrolling monocytes. These cells then patrol the vascular network, engulfing deleterious vascular Aβ in an effort to remove it from the brain. MDP further aids the removal of Aβ from the brain to blood by inducing LRP-1 transports, enabling patrolling monocytes in the blood to scavenge it. The upper right quadrant depicts the PD-1 inhibitor therapy approach, wherein an anti-PD-1 antibody functions as the PD-1 inhibitor. This strategy enhances the mobilization of monocytes to the brain to engulf the cerebral Aβ, an effect that appears to be precipitated by activated T cells subsequent to PD-1 inhibitor therapy. Interestingly, it is postulated that T cell activation summons blood monocytes to the brain. While an anti-PD-1 antibody causes T cell activation, it also facilitates the migration of monocytes into the brain. However, the capacity of this inhibitor to transform monocytes remains a subject of ongoing investigation. The process is understood to hinge on the interaction between CCR2/CCL2 on monocytes. The bottom panel illustrates the amalgamation of therapies, where the PD-1 inhibitor activates the T cells and enhances the potential of anti-PD-1, possibly under the influence of FOXP3+ Treg cells, in the recruitment of monocytes already converted into a non-classical subset by MDP. Ultimately, this approach also triggers the phagocytosis of Aβ in the brain and its subsequent engulfment within the vascular system. Although not featured in the figure, another prospective strategy involves targeting GSK3β downstream of PD-L1 signaling. Owing to insufficient research on this subject to date, its depiction has been omitted. Further investigations into GSK3β may yield a more comprehensive understanding of its role in monocyte function and Aβ elimination.

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
