# Peer review of "The Synergistic Potential of Combining PD-1/PD-L1 Immune Checkpoint Inhibitors with NOD2 Agonists in Alzheimer’s Disease Treatment"

_ijms, 2023, doi:10.3390/ijms241310905_

Round 1
Reviewer 1 Report
The manuscript “The Synergistic Potential of Combining PD-1/PD-L1 Immune Checkpoint Inhibitors with NOD2 Agonists in Alzheimer's Disease Treatment" unveils a timely and innovative combinatorial therapeutic approach employing MDP for NOD2 activation to direct monocytes towards Aβ engulfment via a sink effect, concomitant with the utilization of GSK3β or PD-1/PD-L1 inhibitors to attenuate activated PD-L1 upon NOD2 induction in response to MDP treatment.
Alzheimer's disease (AD) is the most common cause of dementia in elderly people, affecting more than 45 million people worldwide (Association, 2021). AD features progressive memory loss and general cognitive decline. Only a small proportion of cases represent a familial form of AD. The mechanism that controls the chronic progression of AD remains unclear, and effective methods of treatment are still lacking. AD is characterized by extracellular deposition of β-amyloid (Aβ) peptides, intracellular neurofibrillary tau-clusters, and neuroinflammation which are linked to synapse and neuronal loss.
Despite the positive impression of the work, there are a number of minor comments and issues that could improve the manuscript:
1. A little introductory information to update the topic of the review: about the pathogenesis of Alzheimer's disease, the theory of AD, including the theory associated with the deposition of amyloid-beta (Ab).
2. Aging-related cellular and molecular processes including low-grade inflammation are major players in the pathogenesis of cardiovascular disease (CVD) and Alzheimer’s disease (AD). Epidemiological studies report an independent interaction between the development of dementia and the incidence of CVD in several populations, suggesting the presence of overlapping molecular mechanisms. Accumulating experimental and clinical evidence suggests that amyloid-beta (Ab) peptides may function as a link among aging, CVD, and AD. Aging-related vascular and cardiac deposition of Ab induces tissue inflammation and organ dysfunction, both important components of the Alzheimer’s disease amyloid hypothesis.
For example,
https://www.sciencedirect.com/science/article/pii/S0735109720300632?via%3Dihub
In this review, the authors describe the determinants of Ab metabolism, summarize the effects of Ab on atherothrombosis and cardiac dysfunction, discuss the clinical value of Ab1-40 in CVD prognosis and patient risk stratification, and present the therapeutic interventions that may alter Ab metabolism in humans.
Maybe the authors can take this into account in their review. Given this, is it possible to apply their therapeutic approach in patients with cardiovascular diseases?
3. Where the authors refer to their research, it is recommended to give decorated drawings with their own results.
4. Of further importance is the presence a schematic representation comparing NOD2 agonist therapy vs. combination therapy with PD-1 inhibitors The diagram at the end summarizes all the results obtained . This facilitates the perception of the information and allows one to draw synoptic conclusions about the work and enhances the overall impression of the manuscript.
The review has a narrow focus: it focuses on the synergistic potential of combining PD-1/PD-L1 immune checkpoint inhibitors with NOD2 agonists in Alzheimer's disease treatment. Furthermore, the authors say that inhibitors of glycogen synthase kinase 3 beta (GSK3β), which modulates PD-1, exhibit significant potential as AD treatment modalities.
Therefore, readers will expect to see specific mechanisms of involvement of these molecules in the form of signaling pathways, designed in the form of a picture.
Therefore, the review should be supplemented with figures summarizing and illustrating a separate chapter of the review. Without illustrative material, the text is difficult to perceive.
The authors were able to make a significant contribution to the development of effective clinical practices that reduce the burden in patients with AD.
I recommend this manuscript for publication, after implementing the suggested improvements.
Reviewer 2 Report
The Synergistic Potential of Combining PD-1/PD-L1 Immune 2 Checkpoint Inhibitors with NOD2 Agonists in Alzheimer's 3 Disease Treatment
General Comments: The manuscript in general discusses about the conversion of the pro-inflammatory monocytes in the patrolling monocytes by the help of NOD2 receptor agonists. The manuscript then further discusses about the role of patrolling monocytes in clearing the Aβ peptides from the Alzheimers afflicted cereberovascular system. However, authors have also discussed that NOD2 causes PD-1 induction with the help of PD-L1 or GSK3Beta which can interfere with the recruitment of the monocytes to the Abeta plaques and in their removal from the system. Finally, the authors have stated that using PD-L1 or GSK3 beta inhibitors alongwith NOD2 receptors agonists may help the monocytes for recruitment to the Abeta plaques and their removal from the system.
The manuscript in general is very well written and discussed and can be accepted in this current format with some small changes which I recommended below. I wish the authors all the very best.
Queries:
Q1. However, I would like authors to further discriminate between the patrolling monocytes and the microglia in their role in removing the Abeta plqaues from the AD cerebrovascular system. Because their discussion seems to be confusing for the readers?
Q2. Line 162: Sentence is repeating. Please check.
Q3. Line 168: It should be autoimmune diseases
Q4. Line 172: “When a immune cell occurs” sentence incomplete.
Well written overall with some minor corrections at few places which I listed in my review.
Round 2
Reviewer 1 Report
Thank you so much for us point-by-point response to reviewers’ comments.
I think, the authors provided detailed answers to the reviewer's criticism and made all the necessary changes to the manuscript. I recommend this manuscript for publication in present form.